# Constructing the ADHD Child in Historical Children's Literature

**Xiaoyu Hou**

Department of English Literature, School of Literature and Languages, University of Reading, Reading RG6 6EL, UK; xiaoyu.hou@student.reading.ac.uk

**Abstract:** In this article, debates around ideas of childhood and disability will be engaged through the close reading of the retrospective diagnosis of a child with ADHD in an early work of German children's literature (also widely translated, including into English in 1848), Heinrich Hoffmann's poem *Struwwelpeter*. ADHD is one of the most widely diagnosed and medicated childhood developmental disorders of the present day. At the same time, recent debates have raised questions about the diagnostic criteria, the potential side effects and efficacy of medication, and the impact of the current political context on the diagnosis and treatment of ADHD. Drawing on Michel Foucault's classic arguments about bio-power (2008), as well as the most recent work of critical psychology on childhood developmental disorders, the article draws out both how retrospective diagnoses of ADHD and other disorders, including Autistic Spectrum Disorder (ASD), are defined by current criteria within the political context of the current psychological, cultural, and medical controversies.

**Keywords:** Children's Literature; ADHD; disability; childhood; politics; biopower; *Struwwelpeter*





Studies of ADHD regularly cite Heinrich Hoffman's 1844 German children's poem *Struwwelpeter* as the first instance of ADHD in the historical literature. At the same time, these studies rarely actually read the poem itself in any detail to consider how or why this can help to understand diagnoses such as ADHD in the current time. In doing so, I will be following Laurence and McCallum's argument that:

> we seek historical explanation for the possibility of the diagnosis of conduct disorder in something other than psychiatry's explicit refusal to ground its explanation in anything other than the pathological motivations of the individual. But we also seek explanation in something other than the motivations of a centralized and repressive mandate of power. We take as our cue Michel Foucault's suggestion that the pursuit of motivations behind a phenomenon, whether construed at the level of the individual, or in terms of a broader mandate 'from above', leaves aside questions about the actual 'know-how'—the tools taken up by various governing agencies in achieving specific aims. (Laurence and McCallum 2010)

To embark on this close exploration of the children's poem itself in this frame, I will first provide a brief introduction to recent claims about ADHD and why engaging with it is particularly urgent: ADHD is one of the most widely diagnosed and medicated childhood developmental disorders of the present day:

> The estimated number of children aged 3–17 years ever diagnosed with ADHD, according to a national survey of parents, is 6 million (9.8%) using data from 2016–2019. This number includes: 3–5 years: 265,000 (2%), 6–11 years 2.4 million (10%), 12–17 years: 3.3 million (13%). Boys (13%) are more likely to be diagnosed with ADHD than girls (6%). Black, non-Hispanic children and White, non-Hispanic children are more often diagnosed with ADHD (12% and 10%, respectively), than Hispanic children (8%) or Asian, non-Hispanic children (3%).[1] (Centers for Disease Control and Prevention 2022)

At the same time, recent debates have raised questions about the diagnostic criteria, the potential side effects and efficacy of medication, and the impact of the current political context on the diagnosis and treatment of ADHD. Even several of the founding medical and psychological experts on the diagnosis of ADHD have, in recent years, questioned the high rates of diagnosis and treatment:

> The alarmed critics of the Ritalin disaster are now getting support from an entirely different side. The German weekly *Der Spiegel* quoted in its cover story on 2 February 2012 the US American psychiatrist Leon Eisenberg, born in 1922 as the son of Russian Jewish immigrants, who was the 'scientific father of ADHD' and who said at the age of 87, seven months before his death in his last interview: 'ADHD is a prime example of a fictitious disease.'[2] (Belsham 2013)

As Gascon et al. (2022) describe, 'very polarized reactions from health professionals, the media and the public have been observed: some believe that the condition is now better identified or might even still be under-diagnosed, while others consider it to be over-diagnosed.' As an extension of these questions, the current political context has also been considered as an important contributory factor to the place of ADHD in neo-liberal biopolitics: as explored, for instance, in the Netflix documentary 'Take Your Pills' (2018), ADHD diagnoses and medications (methylphenidate[3] and related nervous-system stimulants) (Edwards 2021) are considered as key to studying successfully or operating successfully in high-pressured, long-hour jobs such as in banking or the financial world. As Ott Puumeister (2014) argues, the French historian and philosopher Michel Foucault famously defines 'biopolitics' in relation to 'how this disorder [ADHD]—which is definitely not an irrefutable medical fact—is being constructed through surveys and biopolitical techniques and what kind of a subject ADHD treatment hopes to create.' As Timimi and Taylor (2004) classically argued from a psychiatric perspective, 'ways in which this [childhood] immaturity is understood and made meaningful is a fact of culture'. In this article, I want to consider the implications for this controversial and sensitive area by reading closely how ADHD is retrospectively claimed to be diagnosed in the German children's poem *Struwwelpeter* in order to explore how the current debates about ADHD can be understood both to determine such retrospective diagnoses, but also how in turn this can throw further light on the current controversies, reminding diagnosticians that childhood can be read as a changing historical, cultural and social construction with differing norms for emotions, behavior, and morality.

Paul Graves Hammerness explains the 'historical overview' of ADHD by making the following claim:

> Consistent with this, one can trace ADHD back to early nineteenth-century obser-vations of children and how they differed in behavior. Poems from the early 1800s demonstrate that these behavioral differences among children were observed far before there was a common scientific name for the problem. (Hammerness 2008, p. 5)

There has to be 'consisten[cy]' for 'one' to 'trace ADHD back' in which 'this' is known to be not the same as 'ADHD' but could be understood as differences in 'people'[4] in terms of 'pay[ing] attention to a task' or 'sit[ting] still' that are known from *intuition* and by *comparison*. And 'one' is also known to already have a certain knowledge of 'ADHD' before 'trac[ing]' it 'back', which implies that there is a division between 'one' who 'trace[s]' 'ADHD' 'back' and 'observations of children and how they differed in behavior' that is being 'trace[d]' 'back'. That is to say, 'observations' framed in the 'early nineteenth-century' are being constituted in the present perspective with an already known 'ADHD' through a retrospection on the retrospection. There is then a shift from 'observations' to '[p]oems' in which something is known to be able to 'demonstrate'. In other words, it is the narration that makes a claim about 'behavioral differences among children' to be linked with 'the problem' and knows these 'differences' which could be 'observed' and 'demonstrate[d]' from 'poems' even though there has not been 'a common scientific name' in the 'early 1800s'

yet. This is how, to Hammerness, 'the' known 'problem', which can be read as 'ADHD' that has 'a scientific name', can be 'trace[d]' 'back' to be in relation to 'poems' through the ideas of 'observations' and/or 'demonstrat[ion]'. So Hammerness knows that 'poems' are not relevant to the notion of 'scientific' as is the case for 'ADHD'. But this does not affect the fact that the relation of 'children' in 'poems' to 'ADHD' becomes a certainty of knowledge. At the same time, the 'name for the problem' is known to be multiple with different ways of naming. This is one of many possible names, being 'common' and 'scientific'. Here, 'common' and 'scientific' are known to be different from one another but have to be there together for the naming of 'the problem'. In this way, what constitutes the problem to me is that the 'children' with 'behavioral differences' in 'poems' 'from the early 1800s' can be 'observed' as real and knowable for Hammerness. And these 'children' are known as problematic already on the basis of comparing the present position within the retrospection on the retrospection, despite the fact that their 'problem[s]' (behaviors) are also known not to be 'common' and 'scientific'.

Hammerness also diagnoses the 'children with ADHD' 'to this day' in relation to "Zappelphillip-Syndrom":

> Symptoms of hyperactivity-impulsivity can be found in Phil's behavior, just as we see in children who are diagnosed with ADHD these days. Phil is unable to sit still even for one meal at the table, constantly fidgety, moving about in his chair to the point of falling off it. (Hammerness 2008, p. 6)

For Hammerness, what 'can be found' is 'just as' what 'we see' without deferral, although they are known not to be the same; 'symptoms of hyperactivity-impulsivity' in relation to 'Phil's behavior' which does not happen in 'these days' are different from 'children who are diagnosed with ADHD'. This is how, by making a claim to 'symptoms of hyperactivity-impulsivity', 'Phil' in the poem who is not in 'these days' can be 'diagnosed' as well, even though what 'can be found' of 'Phil's behavior' is not the same case as see[ing] 'children' with 'ADHD' being 'diagnosed'. In addition, there is a certainty of knowledge about 'Phil' being 'unable to sit still' 'even for one meal'. I read this 'even' to be understood as that 'Phil' might be thought of by others to have a possibility to 'sit still' 'for one meal' 'at the table', but it is known that it is impossible for 'Phil' to do that. Thus the narration here claims to know about 'Phil' more and better than others.

This certainty continues to remain by claiming that 'Phil' is 'fidgety'. And 'fidgety' is known to have different kinds: here it is 'constantly' one in terms of 'moving about in his chair'. Moreover, there is 'the point' embedded in this 'moving'. '[F]alling off' is framed to be different from 'moving about' and also can be read as the stop to 'moving about'. The very idea of 'constantly' is, then, constituted paradoxically on the ground of knowing at which 'point' this 'moving about' would stop. In this sense, 'constantly' turns out to have an ending—'falling off it'—which is already known by the narration prior to making a claim to 'Phil' as 'fidgety'. This is how 'constantly' is defined in the service of 'Phil's behavior' to be 'found' as 'fidgety'.

Before discussing further how 'Phil' can be linked with 'children who are diagnosed with ADHD these days' in Hammerness's claim, I want to explain how I read 'Phil' differently in this poem as follows:

> The Story of Fidgety Philip
>
> "Let me see if Philip can
>
> Be a little gentleman;
>
> Let me see if he is able
>
> To sit still for once at table":
>
> Thus Papa bade Phil behave;
>
> And Mamma looked very grave.
>
> But fidgety Phil,
>
> He won't sit still;

He wriggles,

And giggles,

And then, I declare,

Swings backwards and forwards,

And tilts up his chair,

Just like any rocking horse—

"Philip! I am getting cross!" (Hoffmann 1844)

By making a claim to "if", the narration reveals a lack of knowledge about "Philip"—whether or not he could be "a little gentleman", but the narration can know that "Philip" is there available to "see" for "me". And "let me see if" can be understood as an unavailable 'see[ing]' here with a desire to see in one sense. In another sense, being "a little gentleman" could be recognized and examined by 'see[ing]' outside "Philip", which also implies that "Philip" is expected to be "a little gentleman". Then this idea of 'see[ing]' shifts to be in relation to what "a little gentleman" means—being "able [t]o sit still for once at table". At this point, 'see[ing]' can be read to be an ongoing process in order to get to "see" this "sit still" of "he" "for once". The narration knows that "sit still" has not happened yet and claims a self-constructed and self-directed 'see[ing]'. That is to say, there is no one to be there to do the "let[ting]". 'See[ing]' can be produced through which 'Papa', 'Mamma' and 'Phil' are mobilized to react differently to explain the very idea of the necessity of being "a" "gentleman". But there is a difference between the narrative of 'Papa' and 'Mamma' and that of 'Phil': it is not because they are claimed differently as Papa's 'bade' and Mama's 'very grave' look[ing] against 'fidgety Phil'. Instead, 'won't' can be understood as that 'sit still' has not happened yet and cannot come true other than being constituted as 'wriggles' and 'giggles' with which I read this as a conviction—"sit still" will not happen—that comes up before his 'sit[ting]'. So it is not about "sit still" or not. Rather, it is the way 'sit[ting]' is constructed as being not 'still'. This is, as I analyze, all about the narration mobilizing the non/see[ing] of doing differently in which being "a little gentleman" is something that can be relevant to the notion of being civilized and advocated by certain classes of society. The 'fidgety' manner of 'Phil' is framed as the antithesis of being "a" "gentleman", which should be 'bidden' with "getting cross".

See the naughty, restless child

Growing still more rude and wild,

Till his chair falls over quite.

[…]

How Mamma did fret and frown,

When she saw them tumbling down!

And Papa made such a face!

Philip is in sad disgrace. (Hoffmann 1844)

From the above analysis, the claim to "see" can be regarded as an ongoing process being framed in the 'see[ing]' of "me" so as to get to "see" "sit still" "for once". Paradoxically, it is, at the same time, claimed to be known as what is not going to happen. Similarly, what 'the child' does seems like an ongoing 'see[ing]' to the narration on "me" through the claim to '[g]rowing still' here. However, the word '[t]ill' can be read as the certainty of knowledge about the time that it will stop. In this sense, it has a result known to be there instead of waiting to "see" the result by pushing forward with an ongoing 'see[ing]'. In other words, the narration seems to claim that the unavailable 'see[ing]' gradually becomes an available 'see[ing]', compared with the "let me see if" at first. I read that the narration has already known the result—not only did it know this "sit still" had not been achieved yet but also that "sit still" could not be achieved from the beginning. It is the narration on "me" that claims the way in which "if he is able [t]o sit still for once at table" is an unavailable 'see[ing]' temporarily and constitutes it as an ongoing 'see[ing]'. Thus "for once" is known

not to be there from the very beginning. Therefore, there is no unavailable 'see[ing]' of "sit still". As I mentioned above, it is the way in which an available 'see[ing]'/knowing of not "sit still" is constituted as an unavailable 'see[ing]'.

Furthermore, 'a face' can be 'made' by 'Papa' as 'fret and frown' is 'done' by 'Mamma'. Interestingly, both what Mamma and Papa did, respectively, has already happened, whereas the 'sad' of Philip 'in disgrace' is still going on, at this point, in retrospection. That is to say, the claim to 'Philip' 'in sad disgrace' is constructed to last longer than what 'Mamma' and 'Papa' did with regard to 'tumbling down'. As being 'naughty', 'restless', 'rude' and 'wild' is not what "a little gentleman" is supposed to be, all of these need to be included 'in sad disgrace'. In other words, there should be a 'sad disgrace' in being unable to be "a little gentleman" for the 'child'.

> Where is Philip, where is he?
>
> Fairly covered up you see!
>
> Cloth and all are lying on him;
>
> He has pulled down all upon him.
>
> […]
>
> Philip, this is cruel work.
>
> Table all so bare, and ah!
>
> Poor Papa, and poor Mamma
>
> Look quite cross, and wonder how
>
> They shall have their dinner now. (Hoffmann 1844)

Here, 'covered up' can be read as that there is something waiting to be 'see[n]'. And this is the narration making claims to 'you', 'Philip' and 'he' in the see[ing] of "me" in which it knows this 'you' has a possibility to 'see' 'where' 'Philip' 'is' and 'where' 'he' 'is'. So, this is the way in which 'see[ing]' is constituted as 'fairly covered up'. The 'see[ing]' of "me" always shifts, for instance, from 'fidgety' to 'in sad disgrace', then to be in relation to 'cruel work', which implies that "a little gentleman" cannot do 'cruel work' as such. When it comes to 'Papa' and 'Mama', they are 'poor', affected by this 'cruel work', for not having both "a little gentleman" and 'dinner' in expectation. And it is 'their dinner' that is constituted by excluding 'Philip' from 'dinner' at this point. That is to say, 'see[ing]' is constructed as 'fairly covered up', then ends up with the absence of 'Philip' as unseen.

Unlike my understanding of the poem—how 'see[ing]' is produced and positioned differently to explain why being "a little gentleman" matters for 'Philip', Hammerness extends his reading of 'Phil' ongoingly to 'children with ADHD to this day' through a repeated engagement with the issue of 'describe':

> Parents of children with ADHD to this day will often describe their child's level of activity as nonstop for the majority of the day, even when the child is supposed to be doing something as simple as sitting in a chair. (Hammerness 2008, p. 6)

By taking the idea of 'describe' for granted, the perspective shifts from 'poem' to 'parents' in which it knows there are 'parents of children with ADHD' and knows what these 'parents' have describe[d] and 'will' 'describe' 'to this day'. In other words, for Hammerness, 'their child's level of activity' is knowable before this very act of 'describ[ing]': it is 'their child', but as being neither 'parents' nor 'their child', this unitary 'we'[5] are framed to know that the 'activity' of 'their child' has different level[s], here is a 'nonstop' 'level' 'for the majority of the day'. Furthermore, 'even' claims that a 'child' might be thought of by others as having a possibility of 'sitting in a chair' rather than being 'nonstop', since the 'sitting' is constructed as 'simple'. In this way, what constitutes a 'nonstop' 'level of activity' outside a 'child' turns out to be that 'the child' is not doing what he/she is supposed to do, for instance, which can include the idea of 'simple' in 'sitting in a chair'.

For a further and comparable retrospective reading of ADHD in Nineteenth-century children's poetry, we can turn to the work of Bader et al. as they discuss 'Heinrich Hoffmann' in *The Zappel-Philipp a Historical Example of ADHD Clinics* as follows:

> This story (of Zappel-Philipp) provides an example of the debate of pathologizing ordinary childhood quirks. The question remains open, but the interpretation of Zappel-Philipp as a hyperactive and impulsive boy with ADHD still remains speculative without accurate information about Heinrich Hoffmann's intentions. (Bader et al. 2018)

The narration here claims to know that 'an example' can be 'provide[d]' by the 'story (of Zappel-Philipp)'. At the same time, there is no other 'debate' in this 'example' which is in relation to 'pathologizing'. And 'childhood' is known to have 'quirks' of different kinds. It could imply that 'childhood quirks' are also known to be non-'ordinary'. So this very notion of 'debate' is constituted to be in relation to the difference between 'pathologizing' 'childhood' as it is being claimed to have 'quirks' and not 'pathologizing' 'childhood' as 'quirks' can be known to be 'ordinary'. Nevertheless, no matter what the status of 'pathologizing' in 'the debate', 'childhood' is already known to have 'quirks'. Only the question of whether 'childhood quirks' are 'ordinary' remains to be uncertain.

With the narration's shifts from 'childhood quirks' to 'a hyperactive and impulsive boy with ADHD', [t]he 'question' of 'pathologizing' or not is framed as 'open'. The claim to 'interpretation' implies that 'Zappel-Philipp' is not the same as 'a hyperactive and impulsive boy with ADHD' but could be 'interpret[ed]' 'as' that. Simultaneously, 'ADHD' is known to be different from 'the question' being 'open', with the multiple 'remains' being constituted differently: the first 'remains' can be understood to claim that 'the question' is known to be 'open' to 'debate' without the certainty of knowledge with respect to 'pathologizing'. The second 'remains', however, implies the very act of 'interpretation' is known to be 'speculative'. And here is the 'still remains', which implies that non-'speculative' regarding 'Zappel-Philipp' with 'ADHD' has never happened before within the retrospection on the retrospection. In this sense, 'the question' is not something 'speculative' and can be 'open' to 'debate' while the 'interpretation of Zappel-Philipp as a hyperactive and impulsive boy with ADHD' is known as 'speculative' already without needing to be 'open' to 'debate'. Furthermore, the claim to 'speculative' is constructed to be relevant to 'accurate information about Heinrich Hoffmann's intentions' in terms of 'Zappel-Philipp'. In other words, for Bader et al., there could or should be 'Heinrich Hoffmann's intentions' in writing the 'story' of 'Zappel-Philipp', and they know that these 'intentions' can be divided into 'accurate' and inaccurate 'information' which has something to do with the 'interpretation'. So this is how 'the interpretation of Zappel-Philipp as a hyperactive and impulsive boy with ADHD' is known to be 'speculative' due to the very idea of 'without accurate information' being claimed. Therefore, on this score, the narration both knows and does not know at the same time what 'Heinrich Hoffmann's intentions' are regarding the 'story' of 'Zappel-Philipp'.

In the footnotes, Bader, Tannock, and Hadjikhani explain more about 'Heinrich Hoffmann's intentions':

> The historic sources at arrangement do not include specific information with regard to this situation, nor reflections concerning the concept of ADHD. Heinrich Hoffmann [sic] clinical descriptions were destroyed during the fire in the archives of Frankfurt am Main in 1945. (Bader et al. 2018)

Non-'historic' 'sources' regarding 'Heinrich Hoffmann' are also known to exist according to the narration, and, with the claim to 'at', it also knows 'historic' 'sources' include many other things besides 'arrangement'. 'The historic sources at arrangement' are thought to be able to 'include specific information', but it turns out that 'this situation' is not 'includ[ed]'. I read that 'this situation' can be seen as 'the interpretation of Zappel-Philipp as a hyperactive and impulsive boy with ADHD'. Meanwhile, the narration knows what 'reflections' are in relation to 'the concept of ADHD'. But, at this stage, it is something other

than 'the concept of ADHD' that is known as 'reflections' framed in 'arrangement', which can, then, be regarded as 'historic sources'.

When this 'concept of ADHD' is found to fail to be guaranteed by 'historic sources', what 'the concept of ADHD' is supposed to be reflected upon is further discussed in relation to 'Heinrich Hoffmann [sic] clinical descriptions': 'Heinrich Hoffmann' is known to have 'clinical descriptions' which have been 'destroyed'. In this respect, the narration claims a lack of knowledge about 'Heinrich Hoffmann [sic] clinical descriptions'. Nevertheless, this known 'destroyed' does not affect its certainty of knowledge about 'Heinrich Hoffmann's intentions' as analyzed above, nor does it affect the narrations making a claim about 'the interpretation of Zappel-Philipp as a hyperactive and impulsive boy with ADHD' as being 'speculative', as the mysterious 'clinical descriptions' are, after all, regarded and held as an authority and as being 'accurate information' that could be somehow a more reliable source for the poetic creation. So, by searching around 'historic sources at arrangement' as well as firmly believing 'clinical descriptions' regarding 'Heinrich Hoffmann', Bader et al. know what 'accurate information about Heinrich Hoffmann's intentions' is—it is the way 'Heinrich Hoffmann's intentions' are constituted to rely on the known 'historic sources at arrangement' and on the claim about 'clinical descriptions'. So the very notion of non-'speculative' with respect to 'interpretation' implies something should be constructed on the basis of known 'specific information' and/or known 'reflections concerning the concept of ADHD' from 'historic sources at arrangement'. Otherwise, it would be claimed as being 'speculative' no matter when, where, and how 'Heinrich Hoffmann [sic] clinical descriptions' were 'destroyed'. In this sense, the narration knows about 'accurate information' with respect to 'Heinrich Hoffmann's intentions' from known 'historic sources at arrangement' to some extent, without necessarily knowing what 'Heinrich Hoffmann [sic] clinical descriptions' were.

Bader et al. also aspire in the section on 'Heinrich Hoffmann: biographical aspects' to see how 'children's stories' are 'creat[ed]':

> Finding no gift that would please him (Heinrich Hoffmann's son Carl Philipp), he suddenly had the idea of creating children's stories illustrated with colored drawings about children who misbehaved and whose actions subsequently lead to very negative repercussions. He worked on this project, which fulfilled some inner need, during his free time in a spontaneous manner, with little reflection on their content. ([Bader et al. 2018](#))

The narration of 'he' knows that there has to be a 'gift' to be 'found' for pleas[ing] the 'son', and the result of '[f]inding' 'gift' is known as 'no'. There is then a shift from [f]inding 'no gift' to 'had the idea' in which no other idea is known to be relevant to 'creating children's stories illustrated with colored drawings'. And this is 'the idea' that has not been 'had' before but could 'had' 'suddenly' to be in line with what is meant by the claim to both 'a spontaneous manner' and 'little reflection on their content'. Here, 'had the idea' is known to be different from [f]inding a 'gift' but that it could 'please' 'him' as well. And this is the way 'stories' are claimed as belonging to 'children' but being 'creat[ed]' by those who are not 'children' themselves. The narration of the 'idea' of 'he' claims a certainty of knowledge about 'children's stories' in which 'children' are 'creat[ed]' as 'misbehaved'. I read 'whose actions' could be regarded as being in relation to 'misbehaved' 'children' in which even though 'actions' are known to be different from 'misbehaved': these 'actions' are known to 'lead to' 'very negative' 'repercussions' with which the idea of 'subsequently' is constructed as certain. So the narration knows the 'very negative' 'repercussions' will be 'led' by 'actions' in relation to 'creat[ed]' 'children' in the 'stories' before this very notion of 'lead to' happens and be 'subsequently' 'pleas[ing]' the 'son'. Not only are 'children' constituted as 'misbehave[rs]', but also, the 'son' is framed to be someone who would be 'please[d]' by seeing 'children' with the 'very negative repercussions' in the 'stories'.

In addition, the narration of 'he' claims to know about the 'need[s]' of 'he' of different kinds: this is 'some' 'inner' 'need', not all of them, which can be 'fulfilled' by work[ing] 'on this project'. So, there are other things that could 'fulfill' the 'inner need' of 'he'. The

narration also knows all of 'his' 'time' in order to claim that 'worked on this project' is 'during his free time'. Furthermore, 'this project' is known to have been 'worked on' 'in a spontaneous manner'. Specifically, there are different 'manner[s]': this is one of many possible 'spontaneous' 'manner[s]' which can have 'some inner need' 'fulfilled'. 'Reflection' in relation to 'worked on this project', however, is known as 'little', which implies there could or should be 'reflection' regarding 'their content' during 'his' 'work[ing]'. Therefore, this is how 'some inner need', 'a spontaneous manner', and a 'little reflection on their content' are framed in the middle of what the claim to 'worked on this project' means. This is also how 'children's stories' are known to be 'creat[ed]' with the idea of 'a spontaneous manner' and how 'their content' in relation to 'children who misbehaved and whose actions subsequently lead to very negative repercussions' is known 'with little reflection'. From this, I read that 'please him', then, at once, can and cannot be included in this 'some inner need'.

In the section on 'How did Heinrich Hoffman come to be knowledgeable about ADHD?', Bader et al. continue to discuss the idea of 'intentions':

> While Hoffmann gave no precise information on the context of the creation of the characters [sic] *Struwwelpeter*, he mentioned briefly in his autobiography: "These stories are not invented from scratch, one or the other these stories grew up in a fertile soil." We presume that privacy and respect for personal life were the reasons for the lack of precision about the child—or the children—who gave him the idea for this and others [sic] stories. (Bader et al. 2018)

According to the narration, there has to be 'the context' for 'the creation of the characters [sic] *Struwwelpeter*' in which this 'context' is known to have 'information' that can be 'precise' and/or not. And it claims the lack of knowledge about 'precise information' 'on the context of the creation', which implies there can and should be 'precise information' 'give[n]' by 'Hoffmann'. There is something being 'mentioned' to different extents 'in his autobiography'. '[M]entioned' here is known as 'briefly', compared with other things being 'mentioned'. In this way, the claim to 'gave' is constituted on the ground of presuming that 'information' regarding 'the context of the creation' should be 'mentioned' 'in his autobiography'. And why 'information' is defined as not 'precise' is because the narration knows 'information on the context of the creation of the characters [sic] *Struwwelpeter*' is 'mentioned' 'briefly' 'in his autobiography'. Specifically, the ideas about "invented" "stories" are 'mentioned' 'in his autobiography': what "stories" are "invented from scratch" is also known by the narration. But "stories" here which could be read as being in relation to '*Struwwelpeter*' are claimed as being "not" "invented from scratch"; the narration knows where "these stories" "are" "from" instead of 'invent[ing]' them "from scratch". There is no certainty about whether "one" "or" "the other these stories": it is possible for either to "grew up in a fertile soil". This is "a" "fertile soil"—not 'the'—which implies there is other "soil" that could have "these stories" to "grew up" as long as "soil" is known to be "fertile", even though "stories" and "soil" are known to differ from each other. In addition, I read "these stories" of "one" or "the other" are not the same as "these stories" being claimed as "not invented from scratch" since this is a past of a known "grew up" in relation to the idea of "fertile" that is constituted in the present perspective which claims the certain knowledge of what is "from scratch" and what is not within the retrospection. In this way, "stories" are constructed to be "fertile" prior to the very act of "invented". At the same time, by making a claim to "grew up", the "stories" regarding '*Struwwelpeter*' are based on but not the same as "these stories" known before.

Then the narration on and of 'we' claims that there has to be 'the reasons for the lack of precision', which is framed within the idea of 'presume'. So, for Bader et al., there should be an idea of 'presume' when weighing up whether the 'information' for 'the creation of the characters [sic] *Struwwelpeter*' is 'precise' or not; even if the relevant 'information' is 'mentioned' 'briefly' 'in his autobiography', there should be 'the reasons' 'presume[d]' for explaining the 'lack of precision'. To be more specific, here is 'the lack of precision' 'about' 'the child—or the children' that is known to have 'the reasons' that need to be 'presume[d]', which is constituted on the ground of the very ideas of 'privacy' and 'respect'. That is to

say, 'the child—or the children' is already known to be there to 'gave him the idea', while it is 'the reasons for the lack of precision about the child—or the children' which need to be 'presume[d]'. In this sense, 'the child—or the children' is both known and unknown at the same time. In addition, the repetition can always be seen through, for example, 'accurate information', 'clinical descriptions', 'reflection' on 'content', 'his autobiography', and 'precise information'; all of these can be read to be constructed as uncertainty which is actually already known with certainty.

Bader et al. also suggest how the 'Zappel-Philipp's story' could be a 'hypothesis' in relation to issues about 'metaphor', 'translation', and 'image':

> Alongside the open question regarding observable behavior of an ADHD boy, the French translation by Cavanna intuitively uses the metaphor of a "devil in the flesh" to describe the inner movement that goes beyond the child's control capabilities. This metaphor provides an opening to the inner experience of these children and is an incarnation of the lack of self-control of the body, which is a major feature of this syndrome. This image of a "devil in the flesh" also corresponds with ADHD questionnaires [sic] criteria such as "seems always 'under tension', 'as if he worked on batteries'." The Zappel-Philipp's story could represent a historic reference source of the development of the concept of ADHD, but this hypothesis remains speculative. (Bader et al. 2018)

The narration of 'Cavanna' claims that 'the French translation' can be 'intuitively' relevant to 'an ADHD boy' who is known to have 'observable behavior' framed within 'the open question'. In other words, 'an' 'ADHD boy', as one of many possible known 'ADHD boy[s]', is claimed to have 'behavior' which could be 'observable'. But this very idea of 'observable' is 'the' 'question' known to be 'open' to the narration. The narration also knows about 'a "devil in the flesh"' which is translated from 'the French translation' as 'the metaphor'. And, this 'metaphor' is being use[d] 'intuitively' by 'Cavanna' to 'describe' something in relation to 'child's control capabilities'. Therefore, 'a "devil in the flesh"' is known to be different from the 'child' but could be use[d] as 'the metaphor' 'intuitively' to 'describe' this 'child'. To be more specific, 'child' is known to have 'the inner movement' and 'control capabilities' to different extents, respectively. When 'the inner movement' is constituted as 'go[ing]' 'beyond' the 'control capabilities', the 'child' is, then, being describe[d] as 'a "devil in the flesh"' in the name of 'metaphor', which is known as 'intuitive'. And this very idea of 'intuiti[on]' is constructed on the basis of the translation of 'the French translation'. This is how the ideas of 'intuitively', 'metaphor', and 'translation' are mobilized in the service of 'describing' 'the inner movement' and 'control capacities' of 'the child'.

In addition, 'these children' have 'the inner experience', which is known prior to the claim to 'an' 'opening' that can be 'provide[d]' by [t]his 'metaphor'. In one sense, this is 'an' 'opening'—not 'the'—which means there are other opening[s] 'to the inner experience of these children'. In another sense, the claim implies the accessibility of 'the inner experience' as already known to be there before the very notion of 'metaphor'. At the same time, 'the body' can be known as having or 'lack of' 'self-control'. And 'the lack of self-control of the body' is known to have 'incarnation[s]': 'this metaphor' is one of many possible incarnation[s] but not the same as 'the lack of self-control of the body'; 'the lack of self-control of the body' can be only known in its 'incarnation'; at this point, through [t]his 'metaphor'—'a "devil in the flesh"'. Therefore, 'the lack of self-control of the body' is also not the same as 'a "devil in the flesh"' but can only be known as such, which also implies, by reading the 'translation' of 'Cavanna', that this is how Bader et al. understand what 'a devil' means—'the lack of self-control of the body'. According to the narration, it knows that 'this syndrome' has many 'major' 'feature[s]', one of which is relevant to 'the lack of self-control of the body'. This is how the relation between 'an ADHD boy' and a 'child' is connected through 'metaphor' and 'translation' which are known as an 'intuitive' 'use', although 'the lack of self-control of the body' is already known to be different from one of the 'feature[s]' of 'ADHD' by the self-contradictory claim to 'incarnation'.

The division goes on with the claims shifting from 'metaphor' to 'image': at this point, 'a "devil in the flesh"' is claimed to have an 'image' that is, again, known to be different from but able to 'correspond with' 'ADHD questionnaires [sic] criteria'. In this way, the very idea of 'corresponds with' assumes both that 'a "devil in the flesh"' can be framed as and known through the 'image' and that 'ADHD questionnaires [sic] criteria' are knowable. To be more specific, both "under tension" or "worked on batteries" are subject to the claims of "seems" and "as if". This is also how the notions of "seems" and/or "as if" are embedded in one of the many 'questionnaires [sic] criteria' which are constituted in relation to 'ADHD'. In this sense, what an 'image of 'a "devil in the flesh"' means can be only known by but not the same as "under tension" and "worked on batteries" which are constructed only as a "seems" and/or "as if".

'The Zappel-Philipp's story' is known to be different from but 'could' 'represent' one of many possible 'historic reference' 'source [s]' regarding 'the development of the concept of ADHD', as both a 'metaphor' and an 'image' of 'a "devil in the flesh"' are claimed to be in relation to 'ADHD', which is also—as I have analyzed before—how 'historic source[s]' can be produced, developed as such and then taken for granted. At the same time, this possibility of 'represent[ing]' is known as a 'hypothesis' that has not yet become non-'speculative'. So this is how 'Zappel-Philipp's story' regarding the translation of 'the French translation' is constructed to be relevant to 'ADHD' on the basis of the 'hypothesis' known by the narration as being 'speculative' from the beginning, whereas, in terms of 'a historic source', the 'Zappel-Philipp's story' can be a 'represent[ation]' anyway with no need to 'speculate'.

Looking closely at the language of the medical-psychological claims about ADHD in relation to *Struwwelpeter*, then, allows for an analysis that reveals specific investments in ideas of ADHD (and related developmental disabilities or mental illnesses, including ASD (Santa Maria 2015) as an already known set of behaviors or 'inner experience' which correspond to contemporary diagnostic criteria cast back into the historical past. At the same time, this also allows for a consideration of how the contemporary diagnostic criteria themselves also vary and rely in turn on the same strategies of 'observations' which divide behaviors, emotions, and experiences into pathological and non-pathological categories ('quirk'). As Michel Foucault classically argues:

> once again, the intellectual instrument, the type of calculation or form of rationality that made possible the self-limitation of governmental reason was not the law. What is it, starting from the middle of the eighteenth century? Obviously, it is political economy. (Foucault 2008, p. 13)

In other words, the organization of diagnostic categories such as ADHD is shaped by the biopolitical pressures in terms of understandings of what desirable and productive individuals are seen to be in the service of neo-liberal consumer capitalism, where the capacity of workers to comply with the strictures of the working place demands conformity to standards which extend beyond those working places into the very inner lives of those who have no choice but to participate in these systems. Contemporary diagnosticians may usefully keep these issues in mind whilst determining the ways these various factors feed into their differential diagnoses for ADHD and other childhood developmental diagnoses. As the views on ADHD and its treatments are already shifting, it may be that in time other diagnostic categories and treatment options will come to take their place (at least in part), as indeed happened in the past in terms of the diagnoses which ADHD itself replaced.

**Funding:** This research received no external funding.

**Institutional Review Board Statement:** Not applicable.

**Informed Consent Statement:** Not applicable.

**Data Availability Statement:** Not applicable.

**Acknowledgments:** I wish to express my gratitude to Karín Lesnik-Oberstein for her love and support, as well as to Neil Cocks, Sue Walsh, and my colleagues who have read and discussed numerous texts with me over the past few years, which have helped me a lot in my research.

**Conflicts of Interest:** The author declares no conflict of interest.

## Notes

1.     The statistics about ADHD are obtained from Centers for Disease Control and Prevention: https://www.cdc.gov/ncbddd/adhd/data.html (accessed on 14 October 2022).
2.     See Belsham (2013), Inventor of ADHD's Deathbed Confession: "ADHD is a Fictitious Disease".
3.     I take the term from: Edwards (2021), How Effective is Medication for ADHD Symptoms in Children with ASD?
4.     Discussed prior to the quotation here in Hammerness's 'historical overview': 'When thinking about the complex nature of the human brain and the basic differences between people [...] it makes intuitive sense that there would also be differences in individuals' abilities to pay attention, to think and act, at the most basic level. It makes sense that there have always been people who cannot pay attention to a task as long as others or who could not sit still as well as others'.
5.     Mentioned in the quotation before my reading of the poem as 'just as we see in children who are diagnosed with ADHD these days [...]'.

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
