# Peer review of "Constructing the ADHD Child in Historical Children’s Literature"

_humanities, doi:10.3390/h12010003_

Round 1

Reviewer 1 Report

Thank you for the opportunity to read and review this paper. I was intrigued by the focus because this poem gets talked about in literature on ADHD as one of the origins of the scientific description, but I’m not aware of anyone having done much serious analysis of it. It strikes me that this search for origins is part of an attempt to construct a shared image of the past in order to mobilise interests around what is understood to be a contemporary problem that is subject to controversy. In turn this makes me think of whole genres of psychologically oriented literature which tries to do similar things with literary characters (e.g.tigger on the couch’ etc) and first-person retrospective narratives about ‘my adhd and me’. 

Your deconstruction of Hammerness alongside presentation of your own interpretation of ‘Fidgety Phil’ usefully draws out a number of concerns raised by ADHD diagnosis to do with expectations about norms of conduct, but also, and I think this is an important point, about the representative leaps that are necessary to draw direct analogies between this material and the contemporary experiences of children. That there is little acknowledgement of this accompanying the routine use of the poem as a source of legitimacy for the application of scientific knowledge to regulate conduct is a concern worth raising. Neither is there much attention to the partial account of history afforded by this routine use, which you also note.  

One of the broader implications I take from the first half of your analysis concerns the importance of narration for constructing diagnostic cases, in which the production of a legitimate case is the product of a rearrangement of narrative fragments into a ‘solvable problem’ (see Berg, 1992 on the construction of medical disposals, for a sociological take on this).  

Overall a fascinating account, but I would like to see more engagement between your reading of the poem, Foucault, and other relevant Foucaultian literature (you cite Bailey – there is also more on this, Valerie Harwood, Jennifer Laurence, Gordon Tait, Gwynedd Lloyd).

Along with this should be a clearer statement situating yourself among these and providing a bit more focus and direction to your research - what should the reader take from this, what are the implications for research, practice etc..

Author Response

Many thanks to the reviewer for these extremely helpful suggestions. In response, I have made explicit the following:

  • In the introduction, that there is little acknowledgment of [the necessity of representative leaps] accompanying the routine use of the poem as a source of legitimacy for the application of scientific knowledge to regulate conduct is a concern worth raising.
  • Throughout the article: more engagement between my reading of the poem, Foucault, and other relevant Foucaultian literature, including Valerie Harwood and Jennifer Laurence.
  • A clearer statement, both in the introduction and in the conclusion, situating myself among these and providing a bit more focus and direction to my research - what should the reader take from this, what are the implications for research, practice etc..

Reviewer 2 Report

This article presents a detailed close reading of the labelling of a child as having ADHD in a 19th century poem Struwwelpeter (Hoffman 1844). The author carefully analyses the narratorial positions of claims to know ADHD and to know the poet's intentions. The author raises interesting questions about the nature of ADHD and its cultural or organic origins. This article raises questions about whether ADHD is a condition which existed in the 19th century prior to current labels and diagnostic criteria. This is interesting and relevant as it raises questions about current assumptions around the labelling of ADHD and claims around the stability of the category. It would be interesting to connect with articles by Timimi about the social construction of ADHD and also around the variances in cultural and historical attitudes to childhood. It would also be helpful to discuss wider attitudes to childhood related to 'wildness' and civilisation which are touched on here but could be interestingly developed further. How does the retrospective labelling of ADHD overlook general arguments about constructions of childhood for example? Overall, this is an important article which will be of interest to readers with just a little further contextualising in the literature (Timimi) and extrapolation (constructions of childhood).

This article in particular might be helpful:

Timimi, S., & Taylor, E. (2004). ADHD is best understood as a cultural construct. The British Journal of Psychiatry184(1), 8-9.

Author Response

Many thanks to the reviewer for these extremely helpful suggestions. I have responded as follows:

  • Together with my response to reviewer 1’s suggestions of further contextualization in the further literature, I have made sure also to include Timimi.
  • I have included in the introduction and conclusion extrapolation in relation to the constructions of childhood in terms of how the retrospective labeling of ADHD overlooks general arguments about the constructions of childhood.

Reviewer 3 Report

This does address the issue of the use of poetry as the proof of ADHD. This is a clearly heart-felt piece that dismantles the previous research. It is hard to follow at times, given the amount of punctuation.  

Author Response

Many thanks to the reviewer for this helpful reminder: I have tried to clarify and reduce the amount of punctuation.

Round 2

Reviewer 1 Report

Thank you for this revision, I have no further comments.

Reviewer 3 Report

Thank you. The amendments really help to contextualise your argument.